# Detecting Anomalous Event Sequences
# with Temporal Point Processes

**Oleksandr Shchur**[*]
Technical University of Munich
shchur@in.tum.de

**Ali Caner Türkmen**
Amazon Research
atturkm@amazon.com

**Tim Januschowski**
Amazon Research
tjnsch@amazon.com

**Jan Gasthaus**
Amazon Research
gasthaus@amazon.com

**Stephan Günnemann**
Technical University of Munich
guennemann@in.tum.de

## Abstract

Automatically detecting anomalies in event data can provide substantial value in domains such as healthcare, DevOps, and information security. In this paper, we frame the problem of detecting anomalous continuous-time event sequences as out-of-distribution (OoD) detection for temporal point processes (TPPs). First, we show how this problem can be approached using goodness-of-fit (GoF) tests. We then demonstrate the limitations of popular GoF statistics for TPPs and propose a new test that addresses these shortcomings. The proposed method can be combined with various TPP models, such as neural TPPs, and is easy to implement. In our experiments, we show that the proposed statistic excels at both traditional GoF testing, as well as at detecting anomalies in simulated and real-world data.

## 1   Introduction

Event data is abundant in the real world and is encountered in various important applications. For example, transactions in financial systems, server logs, and user activity traces can all naturally be represented as discrete events in continuous time. Detecting anomalies in such data can provide immense industrial value. For example, abnormal entries in system logs may correspond to unnoticed server failures, atypical user activity in computer networks may correspond to intrusions, and irregular patterns in financial systems may correspond to fraud or shifts in the market structure.

Manual inspection of such event data is usually infeasible due to its sheer volume. At the same time, hand-crafted rules quickly become obsolete due to software updates or changing trends (He et al., 2016). Ideally, we would like to have an adaptive system that can learn the normal behavior from the data, and automatically detect abnormal event sequences. Importantly, such a system should detect anomalies in a completely unsupervised way, as high-quality labels are usually hard to obtain.

Assuming "normal" data is available, we can formulate the problem of detecting anomalous event sequences as an instance of out-of-distribution (OoD) detection. Multiple recent works consider OoD detection for image data based on deep generative models (Ren et al., 2019; Nalisnick et al., 2019; Wang et al., 2020). However, none of these papers consider continuous-time event data. Deep generative models for such variable-length event sequences are known as neural temporal point processes (TPPs) (Du et al., 2016). Still, the literature on neural TPPs mostly focuses on prediction tasks, and the problem of anomaly detection has not been adequately addressed by existing works (Shchur et al., 2021). We aim to fill this gap in our paper.

---

[*]Work done during an internship at Amazon Research.
Code and datasets: https://github.com/shchur/tpp-anomaly-detection

35th Conference on Neural Information Processing Systems (NeurIPS 2021).

Our main contributions are the following:

1. **Approach for anomaly detection for TPPs.** We draw connections between OoD detection and GoF testing for TPPs (Section 2). By combining this insight with neural TPPs, we propose an approach for anomaly detection that shows high accuracy on synthetic and real-world event data.

2. **A new test statistic for TPPs.** We highlight the limitations of popular GoF statistics for TPPs and propose the sum-of-squared-spacings statistic that addresses these shortcomings (Section 4). The proposed statistic can be applied to both unmarked and marked TPPs.

## 2   Anomaly detection and goodness-of-fit testing

**Background.** A temporal point process (TPP) (Daley & Vere-Jones, 2003), denoted as $\mathbb{P}$, defines a probability distribution over variable-length event sequences in an interval $[0, T]$. A TPP realization $X$ consists of strictly increasing arrival times $(t_1, \ldots, t_N)$, where $N$, the number of events, is itself a random variable. A TPP is characterized by its conditional intensity function $\lambda^*(t) := \lambda(t|\mathcal{H}_t)$ that is equal to the rate of arrival of new events given the history $\mathcal{H}_t = \{t_j : t_j < t\}$. Equivalently, a TPP can be specified with the integrated intensity function (a.k.a. the compensator) $\Lambda^*(t) = \int_0^t \lambda^*(u) du$.

**Out-of-distribution (OoD) detection.** We formulate the problem of detecting anomalous event sequences as an instance of OoD detection (Liang et al., 2018). Namely, we assume that we are given a large set of training sequences $\mathcal{D}_{\text{train}} = \{X_1, \ldots, X_M\}$ that were sampled i.i.d. from some *unknown* distribution $\mathbb{P}_{\text{data}}$ over a domain $\mathcal{X}$. At test time, we need to determine whether a new sequence $X$ was also drawn from $\mathbb{P}_{\text{data}}$ (i.e., $X$ is in-distribution or "normal") or from another distribution $\mathbb{Q} \neq \mathbb{P}_{\text{data}}$ (i.e., $X$ is out-of-distribution or anomalous). We can phrase this problem as a null hypothesis test:

$$H_0 : X \sim \mathbb{P}_{\text{data}} \qquad H_1 : X \sim \mathbb{Q} \quad \text{for some} \quad \mathbb{Q} \neq \mathbb{P}_{\text{data}}. \qquad (1)$$

To reiterate, here we consider the case where $X$ is a variable-length event sequence and $\mathbb{P}_{\text{data}}$ is some unknown TPP. However, the rest of the discussion in Section 2 also applies to distributions over other data types, such as images.

**Goodness-of-fit (GoF) testing.** First, we observe that the problem of OoD detection is closely related to the problem of GoF testing (D'Agostino, 1986). We now outline the setup and approaches for GoF testing, and then describe how these can be applied to OoD detection. The goal of a GoF test to determine whether a random element $X$ follows a *known* distribution $\mathbb{P}_{\text{model}}$[2]

$$H_0 : X \sim \mathbb{P}_{\text{model}} \quad H_1 : X \sim \mathbb{Q} \quad \text{for some} \quad \mathbb{Q} \neq \mathbb{P}_{\text{model}}. \quad (2)$$

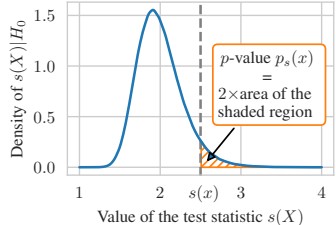

Figure 1: $p$-value is computed as the tail probability under the sampling distribution $s(X)|H_0$.

We can perform such a test by defining a test statistic $s(X)$, where $s \colon \mathcal{X} \to \mathbb{R}$ (Fisher, 1936). For this, we compute the (two-sided) $p$-value for an observed realization $x$ of $X$ as[3]

$$p_s(x) = 2 \times \min\{\Pr(s(X) \le s(x)|H_0), 1 - \Pr(s(X) \le s(x)|H_0)\}. \qquad (3)$$

The factor 2 accounts for the fact that the test is two-sided. We reject the null hypothesis (i.e., conclude that $X$ doesn't follow $\mathbb{P}_{\text{model}}$) if the $p$-value is below some predefined confidence level $\alpha$. Note that computing the $p$-value requires evaluating the cumulative distribution function (CDF) of the sampling distribution, i.e., the distribution test statistic $s(X)$ under the null hypothesis $H_0$.

**GoF testing vs. OoD detection.** The two hypothesis tests (Equations 1 and 2) appear similar—in both cases the goal is to determine whether $X$ follows a certain distribution $\mathbb{P}$ and no assumptions are made about the alternative $\mathbb{Q}$. This means that we can perform OoD detection using the procedure described above, that is, by defining a test statistic $s(X)$ and computing the respective $p$-value (Equation 3). However, in case of GoF testing (Equation 2), the distribution $\mathbb{P}_{\text{model}}$ is known. Therefore, we can analytically compute or approximate the CDF of $s(X)|X \sim \mathbb{P}_{\text{model}}$, and thus the $p$-value. In contrast, in an OoD detection hypothesis test (Equation 1), we make no assumptions about $\mathbb{P}_{\text{data}}$ and only

---

[2]We test a single realization $X$, as is common in TPP literature (Brown et al., 2002). Note that this differs from works on *univariate* GoF testing that consider multiple realizations, i.e., $H_0 : X_1, \ldots, X_M \overset{\text{i.i.d.}}{\sim} \mathbb{P}_{\text{model}}$.

[3]In the rest of the paper, the difference between the random element $X$ and its realization $x$ is unimportant, so we denote both as $X$, as is usually done in the literature.

have access to samples $\mathcal{D}_{\text{train}}$ that were drawn from this distribution. For this reason, we cannot compute the CDF of $s(X)|X \sim \mathbb{P}_{\text{data}}$ analytically. Instead, we can approximate the $p$-value using the empirical distribution function (EDF) of the test statistic $s(X)$ on $\mathcal{D}_{\text{train}}$.

The above procedure can be seen as a generalization of many existing methods for unsupervised OoD detection. These approaches usually define the test statistic based on the log-likelihood (LL) of a generative model fitted to $\mathcal{D}_{\text{train}}$ (Choi et al., 2018; Ren et al., 2019; Ruff et al., 2021). However, as follows from our discussion above, there is no need to limit ourselves to LL-based statistics. For instance, we can define a test statistic for event sequences based on the rich literature on GoF testing for TPPs. We show in Section 6 that this often leads to more accurate anomaly detection compared to LL. Moreover, the difference between OoD detection and GoF testing is often overlooked. By drawing a clear distinction between the two, we can avoid some of the pitfalls encountered by other works (Nalisnick et al., 2019), as we elaborate in Appendix A.

The anomaly detection framework we outlined above can be applied to any type of data—such as images or time series—but in this work we mostly focus on continuous-time event data. This means that our main goal is to find an appropriate test statistic for variable-length continuous-time event sequences. In Section 3, we take a look at existing GoF statistics for TPPs and analyze their limitations. Then in Section 4, we propose a new test statistic that addresses these shortcomings and describe in more detail how it can be used for OoD detection.

## 3 Review of existing GoF test statistics for TPPs

Here, we consider a GoF test (Equation 2), where the goal is to determine whether an event sequence $X = (t_1, \ldots, t_N)$ was generated by a known TPP $\mathbb{P}_{\text{model}}$ with compensator $\Lambda^*$. We will return to the problem of OoD detection, where the data-generating distribution $\mathbb{P}_{\text{data}}$ is unknown, in Section 4.2.

Many popular GoF tests for TPPs are based on the following result (Ogata, 1988; Brown et al., 2002).

**Theorem 1** (Random time change theorem (Brown et al., 2002)). *A sequence $X = (t_1, \ldots, t_N)$ is distributed according to a TPP with compensator $\Lambda^*$ on the interval $[0, V]$ if and only if the sequence $Z = (\Lambda^*(t_1), \ldots, \Lambda^*(t_N))$ is distributed according to the standard Poisson process on $[0, \Lambda^*(V)]$.*

Intuitively, Theorem 1 can be viewed as a TPP analogue of how the CDF of an arbitrary random variable over $\mathbb{R}$ transforms its realizations into samples from $\text{Uniform}([0, 1])$. Similarly, the compensator $\Lambda^*$ converts a random event sequence $X$ into a realization $Z$ of the standard Poisson process (SPP). Therefore, the problem of GoF testing for an arbitrary TPP reduces to testing whether the transformed sequence $Z$ follows the SPP on $[0, \Lambda^*(T)]$. In other words, we can define a GoF statistic for a TPP with compensator $\Lambda^*$ by (1) applying the compensator to $X$ to obtain $Z$ and (2) computing one of the existing GoF statistics for the SPP on the transformed sequence. This can also be generalized to marked TPPs (where events can belong to one of $K$ classes) by simply concatenating the transformed sequences $Z^{(k)}$ for each event type $k \in \{1, \ldots, K\}$ (see Appendix D for details).

SPP, i.e., the Poisson process with constant intensity $\lambda^*(t) = 1$, is the most basic TPP one can conceive. However, as we will shortly see, existing GoF statistics even for this simple model have considerable shortcomings and can only detect a limited class of deviations from the SPP. More importantly, test statistics for general TPPs defined using the above recipe (Theorem 1) inherit the limitations of the SPP statistics.

For brevity, we denote the transformed arrival times as $Z = (v_1, \ldots, v_N) = (\Lambda^*(t_1), \ldots, \Lambda^*(t_N))$ and the length of the transformed interval as $V = \Lambda^*(T)$. One way to describe the generative process of an SPP is as follows (Pasupathy, 2010)

$$N|V \sim \text{Poisson}(V) \qquad u_i|N, V \sim \text{Uniform}([0, V]) \quad \text{for } i = 1, \ldots, N. \qquad (4)$$

An SPP realization $Z = (v_1, \ldots, v_N)$ is obtained by sorting the $u_i$'s in increasing order. This is equivalent to defining the arrival time $v_i$ as the $i$-th order statistic $u_{(i)}$. We can also represent $Z$ by the inter-event times $(w_1, \ldots, w_{N+1})$ where $w_i = v_i - v_{i-1}$, assuming $v_0 = 0$ and $v_{N+1} = V$.

Barnard (1953) proposed a GoF test for the SPP based on the above description (Equation 4) and the Kolmogorov–Smirnov (KS) statistic. The main idea of this approach is to check whether the arrival times $v_1, \ldots, v_N$ are distributed uniformly in the $[0, V]$ interval. For this, we compare $\hat{F}_{\text{arr}}$, the empirical CDF of the arrival times, with $F_{\text{arr}}(u) = u/V$, the CDF of the $\text{Uniform}([0, V])$ distribution.

This can be done using the *KS statistic on the arrival times (KS arrival)*, defined as

$$\kappa_{\text{arr}}(Z) = \sqrt{N} \cdot \sup_{u \in [0,V]} |\hat{F}_{\text{arr}}(u) - F_{\text{arr}}(u)| \qquad \text{where} \qquad \hat{F}_{\text{arr}}(u) = \frac{1}{N} \sum_{i=1}^{N} \mathbb{1}(v_i \leq u). \qquad (5)$$

Another popular GoF test for the SPP is based on the fact that the inter-event times $w_i$ are distributed according to the Exponential(1) distribution (Cox, 1966). The test compares $\hat{F}_{\text{int}}$, the empirical CDF of the inter-event times, and $F_{\text{int}}(u) = 1 - \exp(-u)$, the CDF of the Exponential(1) distribution. This leads to the *KS statistic for the inter-event times (KS inter-event)*

$$\kappa_{\text{int}}(Z) = \sqrt{N} \cdot \sup_{u \in [0,\infty)} |\hat{F}_{\text{int}}(u) - F_{\text{int}}(u)| \quad \text{where} \quad \hat{F}_{\text{int}}(u) = \frac{1}{N+1} \sum_{i=1}^{N+1} \mathbb{1}(w_i \leq u). \qquad (6)$$

KS arrival and KS inter-event statistics are often presented as the go-to approach for testing the goodness-of-fit of the standard Poisson process (Daley & Vere-Jones, 2003). Combining them with Theorem 1 leads to simple GoF tests for arbitrary TPPs that are widely used to this day (Gerhard et al., 2011; Alizadeh et al., 2013; Kim & Whitt, 2014; Tao et al., 2018; Li et al., 2018).

**Limitations of the KS statistics.** The KS statistics $\kappa_{\text{arr}}(Z)$ and $\kappa_{\text{int}}(Z)$ are only able to differentiate the SPP from a narrow class of alternative processes. For example, KS arrival only checks if the arrival times $v_i$ are distributed uniformly, conditioned on the event count $N$. But what if the observed $N$ is itself extremely unlikely under the SPP (Equation 4)? KS inter-event can be similarly insensitive to the event count—removing all events $\frac{V}{2} < v_i \leq V$ from an SPP realization $Z$ will only result in just a single atypically large inter-event time $w_i$, which changes the value of $\kappa_{\text{int}}(Z)$ at most by $\frac{1}{N+1}$. We demonstrate these limitations of $\kappa_{\text{arr}}(Z)$ and $\kappa_{\text{int}}(Z)$ in our experiments in Section 6.1. Other failure modes of the KS statistics were described by Pillow (2009). Note that ad-hoc fixes to the KS statistics do not address these problems. For example, combining multiple tests performed separately for the event count and arrival times using Fisher's method (Fisher, 1948; Cox, 1966) consistently decreases the accuracy, as we show in Appendix G. In the next section, we introduce a different test statistic that aims to address these shortcomings.

## 4 Sum-of-squared-spacings (3S) statistic for TPPs

### 4.1 Goodness-of-fit testing with the 3S statistic

A good test statistic should capture multiple properties of the SPP at once: it should detect deviations w.r.t. both the event count $N$ and the distribution of the arrival or inter-event times. Here, we propose to approach GoF testing with a *sum-of-squared-spacings (3S) statistic* that satisfies these desiderata,

$$\psi(Z) = \frac{1}{V} \sum_{i=1}^{N+1} w_i^2 = \frac{1}{V} \sum_{i=1}^{N+1} (v_i - v_{i-1})^2. \qquad (7)$$

This statistic extends the sum-of-squared-spacings statistic proposed as a test of uniformity for fixed-length samples by Greenwood (1946). The important difference between our definition (Equation 7) and prior works (D'Agostino, 1986) is that we, for the first time, consider the TPP setting, where the number of events $N$ is random as well. For this reason, we use the normalizing constant $1/V$ instead of $N/V^2$ (see Appendix B for details). As we will see, this helps capture abnormalities in the event count and results in more favorable asymptotic properties for the case of SPP.

Intuitively, for a fixed $N$, the statistic $\psi$ is maximized if the spacings are extremely imbalanced, i.e., if one inter-event time $w_i$ is close to $V$ and the rest are close to zero. Conversely, $\psi$ attains its minimum when the spacings are all equal, that is $w_i = \frac{V}{N+1}$ for all $i$.

In Figure 2a we visualize the distribution of $\psi | N, V$ for two different values of $N$. We see that the distribution of $\psi$ depends strongly on $N$, therefore a GoF test involving $\psi$ will detect if the event count $N$ is atypical for the given SPP. This is in contrast to $\kappa_{\text{arr}}$ and $\kappa_{\text{int}}$, the distributions of which, by design, are (asymptotically) invariant under $N$ (Figure 2b). Even if one accounts for this effect, e.g., by removing the correction factor $\sqrt{N}$ in Equations 5 and 6, their distributions change only slightly compared to the sum of squared spacings (see Figures 2c and 2d). To analyze other properties of the statistic, we consider its moments under the null hypothesis.

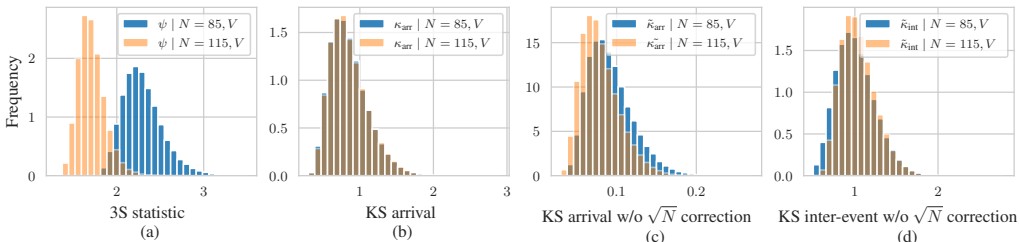

Figure 2: Distribution of different test statistics for the standard Poisson process on $[0, 100]$, conditioned on different event counts $N$. The 3S statistic allows us to differentiate between different values of $N$, while the KS statistics are not sensitive to the changes in $N$.

**Proposition 1.** *Suppose the sequence $Z$ is distributed according to the standard Poisson process on the interval $[0, V]$. Then the first two moments of the statistic $\psi := \psi(Z)$ are*

$$\mathbb{E}[\psi|V] = \frac{2}{V}(V + e^{-V} - 1) \quad and \quad \text{Var}[\psi|V] = \frac{4}{V^2}(2V - 7 + e^{-V}(2V^2 + 4V + 8 - e^{-V})).$$

The proof of Proposition 1 can be found in Appendix C. From Proposition 1 it follows that

$$\lim_{V \to \infty} \mathbb{E}[\psi|V] = 2 \qquad\qquad \lim_{V \to \infty} \text{Var}[\psi|V] = 0. \qquad (8)$$

This leads to a natural notion of *typicality* in the sense of Nalisnick et al. (2019) and Wang et al. (2020) for the standard Poisson process. We can define the typical set of the SPP as the set of variable-length sequences $Z$ on the interval $[0, V]$ that satisfy $|\psi(Z) - 2| \leq \epsilon$ for some small $\epsilon > 0$. It follows from Equation 8 and Chebyshev's inequality that for large enough $V$, the SPP realizations will fall into the typical set with high probability. Therefore, at least for large $V$, we should be able to detect sequences that are not distributed according the SPP based on the statistic $\psi$.

**Summary.** To test the GoF of a TPP with a known compensator $\Lambda^*$ for an event sequence $X = (t_1, \ldots, t_N)$, we first obtain the transformed sequence $Z = (\Lambda^*(t_1), \ldots, \Lambda^*(t_N))$ and compute the statistic $\psi(Z)$ according to Equation 7. Since the CDF of the statistic under $H_0$ cannot be computed analytically, we approximate it using samples drawn from $\mathbb{P}_{\text{model}}$. That is, we draw realizations $\mathcal{D}_{\text{model}} = \{X_1, \ldots, X_M\}$ from the TPP (e.g., using the inversion method (Rasmussen, 2018)) and compute the $p$-value for $X$ (Equation 3) using the EDF of the statistic on $\mathcal{D}_{\text{model}}$ (North et al., 2002).

### 4.2 Out-of-distribution detection with the 3S statistic

We now return to the original problem of OoD detection in TPPs, where we have access to a set of in-distribution sequences $\mathcal{D}_{\text{train}}$ and do not know the data-generating process $\mathbb{P}_{\text{data}}$.

Our idea is to perform the OoD detection hypothesis test (Equation 1) using the sum-of-squared-spacings test statistic that we introduced in the previous section. However, since the data-generating TPP $\mathbb{P}_{\text{data}}$ is unknown, we do not know the corresponding compensator that is necessary to compute the statistic. Instead, we can fit a neural TPP model $\mathbb{P}_{\text{model}}$ (Du et al., 2016) to the sequences in $\mathcal{D}_{\text{train}}$ and use the compensator $\Lambda^*$ of the learned model to compute the statistic $s(X)$.[4] High flexibility of neural TPPs allows these models to more accurately approximate the true compensator. Having defined the statistic, we can approximate its distribution under $H_0$ (i.e., assuming $X \sim \mathbb{P}_{\text{data}}$) by the EDF of the statistic on $\mathcal{D}_{\text{train}}$. We

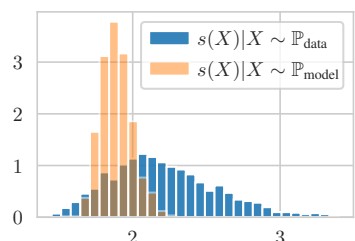

Figure 3: Sampling distribution for the OoD test (blue) and the GoF test (orange). While the same statistic $s(X)$ is used in both cases, the $p$-values are computed differently depending on which test we perform.

use this EDF to compute the $p$-values for our OoD detection hypothesis test and thus detect anomalous sequences. We provide the pseudocode description of our OoD detection method in Appendix D.

We highlight that an OoD detection procedure like the one above is *not* equivalent to a GoF test for the learned generative model $\mathbb{P}_{\text{model}}$, as suggested by earlier works (Nalisnick et al., 2019). While we use

---

[4]We can replace the 3S statistic on the transformed sequence $Z$ with any other statistic for the SPP, such as KS arrival. In Sections 6.2 and 6.3, we compare different statistics constructed this way.

the compensator of the learned model to define the test statistic $s(X)$, we compute the $p$-value for the OoD detection test based on $s(X)|X \sim \mathbb{P}_{\text{data}}$. This is different from the distribution $s(X)|X \sim \mathbb{P}_{\text{model}}$ used in a GoF test, since in general $\mathbb{P}_{\text{model}} \neq \mathbb{P}_{\text{data}}$. Therefore, even if the distribution of a test statistic under the GoF test can be approximated analytically (as, e.g., for the KS statistic (Marsaglia et al., 2003)), we have to use the EDF of the statistic on $\mathcal{D}_{\text{train}}$ for the OoD detection test. Figure 3 visualizes this difference. Here, we fit a TPP model on the in-distribution sequences from the STEAD dataset (Section 6.3) and plot the empirical distribution of the respective statistic $s(X)$ on $\mathcal{D}_{\text{train}}$ (corresponds to $s(X)|X \sim \mathbb{P}_{\text{data}}$) and on model samples $\mathcal{D}_{\text{model}}$ (corresponds to $s(X)|X \sim \mathbb{P}_{\text{model}}$).

## 5 Related work

**Unsupervised OoD detection.** OoD detection approaches based on deep generative models (similar to our approach in Section 4.2) have received a lot of attention in the literature. However, there are several important differences between our method and prior works. First, most existing approaches perform OoD detection based on the log-likelihood (LL) of the model or some derived statistic (Choi et al., 2018; Ren et al., 2019; Nalisnick et al., 2019; Morningstar et al., 2021; Ruff et al., 2021). We observe that LL can be replaced by any other test statistic, e.g., taken from the GoF testing literature, which often leads to more accurate anomaly detection (Section 6). Second, unlike prior works, we draw a clear distinction between OoD detection and GoF testing. While this difference may seem obvious in hindsight, it is not acknowledged by the existing works, which may lead to complications (see Appendix A). Also, our formulation of the OoD detection problem in Section 2 provides an intuitive explanation to the phenomenon of "typicality" (Nalisnick et al., 2019; Wang et al., 2020). The $(\epsilon, 1)$-typical set of a distribution $\mathbb{P}$ simply corresponds to the acceptance region of the respective hypothesis test with confidence level $\epsilon$ (Equation 1). Finally, most existing papers study OoD detection for image data and none consider variable-length event sequences, which is the focus of our work.

Our OoD detection procedure is also related to the *rarity* anomaly score (Ferragut et al., 2012; Janzing et al., 2019). The rarity score can be interpreted as the negative logarithm of a one-sided $p$-value (Equation 3) of a GoF test that uses the log-likelihood of some *known* model as the test statistic. In contrast, we consider a broader class of statistics and learn the model from the data.

**Anomaly detection for TPPs.** OoD detection, as described in Section 2, is not the only way to formalize anomaly detection for TPPs. For example, Ojeda et al. (2019) developed a distance-based approach for Poisson processes. Recently, Zhu et al. (2020) proposed to detect anomalous event sequences with an adversarially-trained model. Unlike these two methods, our approach can be combined with any TPP model without altering the training procedure. Liu & Hauskrecht (2019) studied anomalous *event* detection with TPPs, while we are concerned with entire event sequences.

**GoF tests for TPPs.** Existing GoF tests for the SPP usually check if the arrival times are distributed uniformly, using, e.g., the KS (Lewis, 1965) or chi-squared statistic (Cox, 1955). Our 3S statistic favorably compares to these approaches thanks to its dependence on the event count $N$, as we explain in Section 4 and show experimentally in Section 6.1. Methods combining the random time change theorem with a GoF test for the SPP (usually, the KS test) have been used at least since Ogata (1988), and are especially popular in neuroscience (Brown et al., 2002; Gerhard et al., 2011; Tao et al., 2018). However, these approaches inherit the limitations of the underlying KS statistic. Replacing the KS score with the 3S statistic consistently leads to a better separation between different TPP distributions (Section 6).

Gerhard & Gerstner (2010) discussed several GoF tests for discrete-time TPPs, while we deal with continuous time. Yang et al. (2019) proposed a GoF test for point processes based on Stein's identity, which is related to a more general class of kernel-based GoF tests (Chwialkowski et al., 2016; Liu et al., 2016). Their approach isn't suitable for neural TPPs, where the Papangelou intensity cannot be computed analytically. A recent work by Wei et al. (2021) designed a GoF test for self-exciting processes under model misspecification. In contrast to these approaches, our proposed GoF test from Section 4.1 can be applied to any TPP with a known compensator.

**Sum-of-squared-spacings statistic.** A similar statistic was first used by Greenwood (1946) for testing whether a fixed number of points are distributed uniformly in an interval. Several follow-up works studied the limiting distribution of the statistic (conditioned on $N$) as $N \to \infty$ (Hill, 1979; Stephens, 1981; Rao & Kuo, 1984). Our proposed statistic (Equation 7) is not invariant w.r.t. $N$ and, therefore, is better suited for testing TPPs. We discuss other related statistics in Appendix B.

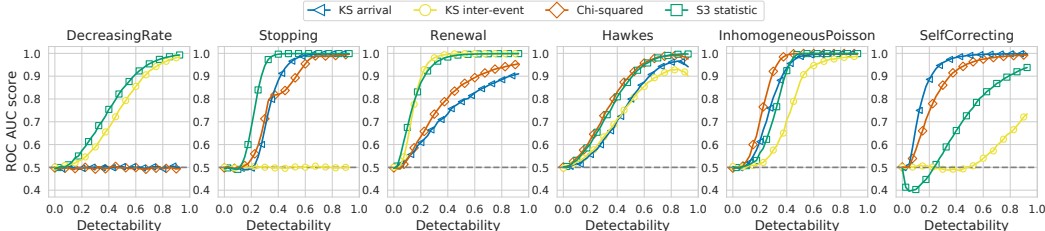

Figure 4: GoF testing for the standard Poisson process using different test statistics, measured with ROC AUC (higher is better). See Section 6.1 for the description of the experimental setup.

# 6 Experiments

Our experimental evaluation covers two main topics. In Section 6.1, we compare the proposed 3S statistic with existing GoF statistics for the SPP. Then in Sections 6.2 and 6.3, we evaluate our OoD detection approach on simulated and real-world data, respectively. The experiments were run on a machine with a 1080Ti GPU. Details on the setup and datasets construction are provided in Appendix E & F.

## 6.1 Standard Poisson process

In Section 3 we mentioned several failure modes of existing GoF statistics for the SPP. Then, in Section 4.1 we introduced the 3S statistic that was supposed to address these limitations. Hence, the goal of this section is to compare the proposed statistic with the existing ones in the task of GoF testing for the SPP. We consider four test statistics: (1) KS statistic on arrival times (Equation 5), (2) KS statistic on inter-event times (Equation 6), (3) chi-squared statistic on the arrival times (Cox, 1955; Tao et al., 2018), and (4) the proposed 3S statistic (Equation 7).

To quantitatively compare the discriminative power of different statistics, we adopt an evaluation strategy similar to Gerhard & Gerstner (2010); Yang et al. (2019). First, we generate a set $\mathcal{D}_{\text{model}}$ consisting of 1000 SPP realizations. We use $\mathcal{D}_{\text{model}}$ to compute the empirical distribution function of each statistic $s(Z)$ under $H_0$. Then, we define two test sets: $\mathcal{D}_{\text{test}}^{\text{ID}}$ (consisting of samples from $\mathbb{P}_{\text{model}}$, the SPP) and $\mathcal{D}_{\text{test}}^{\text{OOD}}$ (consisting of samples from $\mathbb{Q}$, another TPP), each with 1000 sequences. Importantly, in this and following experiments, the training and test sets are always disjoint.

We follow the GoF testing procedure described at the end of Section 4.1, which corresponds to the hypothesis test in Equation 2. That is, we compute the $p$-value (Equation 3) for each sequence in the test sets using the EDF of $s(Z)$ on $\mathcal{D}_{\text{model}}$. A good test statistic $s(Z)$ should assign lower $p$-values to the OoD sequences from $\mathcal{D}_{\text{test}}^{\text{OOD}}$ than to ID sequences from $\mathcal{D}_{\text{test}}^{\text{ID}}$, allowing us to discriminate between samples from $\mathbb{Q}$ and $\mathbb{P}_{\text{model}}$. We quantify how well a given statistic separates the two distributions by computing the area under the ROC curve (ROC AUC). This effectively averages the performance of a statistic for the GoF hypothesis test over different significance levels $\alpha$.

**Datasets.** We consider six choices for the distribution $\mathbb{Q}$:

- RATE, a homogeneous Poisson process with intensity $\mu < 1$;
- STOPPING, where events stop after some time $t_{\text{stop}} \in [0, V]$;
- RENEWAL, where inter-event times are drawn i.i.d. from the Gamma distribution;
- HAWKES, where events are more clustered compared to the SPP;
- INHOMOGENEOUS, a Poisson process with non-constant intensity $\lambda(t) = \beta \sin(\omega t)$;
- SELFCORRECTING, where events are more evenly spaced compared to the SPP.

For cases the last 4 cases, the expected number of events is the same as for the SPP.

For each choice of $\mathbb{Q}$ we define a *detectability* parameter $\delta \in [0, 1]$, where higher $\delta$ corresponds to TPPs that are increasingly dissimilar to the SPP. That is, setting $\delta = 0$ corresponds to a distribution $\mathbb{Q}$ that is exactly equal to the SPP, and $\delta = 1$ corresponds to a distribution that deviates significantly from the SPP. For example, for a Hawkes with conditional intensity $\lambda^*(t) = \mu + \beta \sum_{t_j < t} \exp(-(t - t_j))$, the detectability value of $\delta = 0$ corresponds to $\mu = 1$ and $\beta = 0$ (i.e., $\lambda^*(t) = 1$) making $\mathbb{Q}$

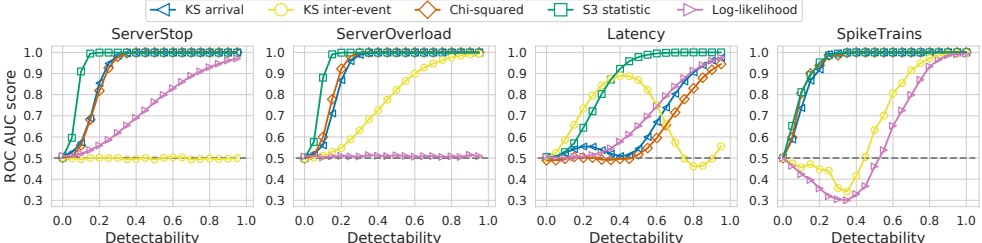

Figure 5: OoD detection on simulated data using different test statistics, measured with with ROC AUC (higher is better). See Section 6.2 for the description of the experimental setup.

indistinguishable from $\mathbb{P}$. The value of $\delta = 0.5$ corresponds to $\mu = 0.5$ and $\beta = 0.5$, which preserves the expected number of events $N$ but makes the arrival times $t_i$ "burstier." We describe how the parameters of each distribution $\mathbb{Q}$ are defined based on $\delta$ in Appendix E. Note that, in general, the ROC AUC scores are not guaranteed to monotonically increase as the detectability $\delta$ is increased.

**Results.** In Figure 4, we present AUC scores for different statistics as $\delta$ is varied. As expected, KS arrival accurately identifies sequences that come from $\mathbb{Q}$ where the absolute time of events are non-uniform (as in INHOMOGENEOUS). Similarly, KS inter-event is good at detecting deviations in the distribution of inter-event times, as in RENEWAL. The performance of the chi-squared statistic is similar to that of KS arrival. Nevertheless, the above statistics fail when the expected number of events, $N$, changes substantially—as in KS arrival and chi-squared on RATE, and KS inter-event on STOPPING. These failure modes match our discussion from Section 3.

In contrast, the 3S statistic stands out as the most consistent test (best or close-to-best performance in 5 out of 6 cases) and does not completely fail in any of the scenarios. The relatively weaker performance on SELFCORRECTING implies that the 3S statistic is less sensitive to superuniform spacings (D'Agostino, 1986) than to imbalanced spacings. The results show that the 3S statistic is able to detect deviations w.r.t. both the event count $N$ (RATE and STOPPING), as well as the distributions of the inter-event times $w_i$ (RENEWAL) or the arrival times $v_i$ (HAWKES and INHOMOGENEOUS)— something that other GoF statistics for the SPP cannot provide.

## 6.2 Detecting anomalies in simulated data

In this section, we test the OoD detection approach discussed in Section 4.2, i.e., we perform anomaly detection for a TPP with an *unknown* compensator. This corresponds to the hypothesis test in Equation 1. We use the training set $\mathcal{D}_{\text{train}}$ to fit an RNN-based neural TPP model (Shchur et al., 2020) via maximum likelihood estimation (see Appendix F for details). Then, we define test statistics for the general TPP as follows. We apply the compensator $\Lambda^*$ of the learned model to each event sequence $X$ and compute the four statistics for the SPP from Section 6.1 on the transformed sequence $Z = \Lambda^*(X)$. We highlight that these methods are not "baselines" in the usual sense—the idea of combining a GoF statistic with a learned TPP model to detect anomalous event sequences is itself novel and hasn't been explored by earlier works. The rest of the setup is similar to Section 6.1. We use $\mathcal{D}_{\text{train}}$ to compute the EDF of each statistic under $H_0$, and then compute the ROC AUC scores on the $p$-values. In addition to the four statistics discussed before, we consider a two-sided test on the log-likelihood $\log q(X)$ of the learned generative model, which corresponds to the approach by Nalisnick et al. (2019).

**Datasets.** Like before, we define a detectability parameter $\delta$ for each scenario that determines how dissimilar ID and OoD sequences are. SERVER-STOP, SERVER-OVERLOAD and LATENCY are inspired by applications in DevOps, such as detecting anomalies in server logs.

- SERVER-OVERLOAD and SERVER-STOP contain data generated by a multivariate Hawkes process with 3 marks, e.g., modeling network traffic among 3 hosts. In OoD sequences, we change the influence matrix to simulate scenarios where a host goes offline (SERVER-STOP), and where a host goes down and the traffic is routed to a different host (SERVER-OVERLOAD). Higher $\delta$ implies that the change in the influence matrix happens earlier.

- LATENCY contains events of two types, sampled as follows. The first mark, the "trigger," is sampled from a homogeneous Poisson process with rate $\mu = 3$. The arrival times of the second

Table 1: ROC AUC scores for OoD detection on real-world datasets (mean & standard error are computed over 5 runs). Best result in **bold**, results within 2 pp. of the best underlined.

| | KS arrival | KS inter-event | Chi-squared | Log-likelihood | 3S statistic |
|---|---|---|---|---|---|
| Logs — Packet corruption (1%) | $57.4 \pm 1.7$ | $62.1 \pm 0.9$ | $66.6 \pm 1.8$ | $75.9 \pm 0.1$ | $\mathbf{95.5} \pm 0.3$ |
| Logs — Packet corruption (10%) | $59.2 \pm 2.3$ | $\underline{97.8} \pm 0.6$ | $59.1 \pm 2.3$ | $\underline{99.0} \pm 0.0$ | $\mathbf{99.4} \pm 0.1$ |
| Logs — Packet duplication (1%) | $81.1 \pm 5.2$ | $82.8 \pm 5.0$ | $74.6 \pm 6.5$ | $88.1 \pm 0.1$ | $\mathbf{90.9} \pm 0.3$ |
| Logs — Packet delay (frontend) | $95.6 \pm 1.2$ | $\underline{98.9} \pm 0.4$ | $\mathbf{99.3} \pm 0.1$ | $90.9 \pm 0.0$ | $\underline{97.6} \pm 0.1$ |
| Logs — Packet delay (all services) | $\mathbf{99.8} \pm 0.0$ | $94.7 \pm 1.1$ | $\mathbf{99.8} \pm 0.0$ | $96.1 \pm 0.0$ | $\underline{99.6} \pm 0.1$ |
| STEAD — Anchorage, AK | $59.6 \pm 0.2$ | $79.7 \pm 0.1$ | $67.4 \pm 0.2$ | $\underline{88.0} \pm 0.1$ | $\mathbf{88.3} \pm 0.6$ |
| STEAD — Aleutian Islands, AK | $53.8 \pm 0.5$ | $88.8 \pm 0.3$ | $62.2 \pm 0.9$ | $\underline{97.0} \pm 0.0$ | $\mathbf{99.8} \pm 0.0$ |
| STEAD — Helmet, CA | $59.1 \pm 0.9$ | $\mathbf{98.7} \pm 0.0$ | $70.0 \pm 0.6$ | $\underline{96.9} \pm 0.0$ | $92.6 \pm 0.3$ |

mark, the "response," are obtained by shifting the times of the first mark by an offset sampled i.i.d. from $\mathrm{Normal}(\mu = 1, \sigma = 0.1)$. In OoD sequences, the delay is increased by an amount proportional to $\delta$, which emulates an increased latency in the system.

- SPIKETRAINS (Stetter et al., 2012) contains sequences of firing times of 50 neurons, each represented by a distinct mark. We generate OoD sequences by shuffling the indices of $k$ neurons (e.g., switching marks 1 and 2), where higher detectability $\delta$ implies more switches $k$. Here we study how different statistics behave for TPPs with a large number of marks.

**Results** are shown in Figure 5. The 3S statistic demonstrates excellent performance in all four scenarios, followed by KS arrival and chi-squared. In case of SERVER-STOP and SERVER-OVERLOAD, the 3S statistic allows us to perfectly detect the anomalies even when only 5% of the time interval are affected by the change in the influence structure. KS inter-event and log-likelihood statistics completely fail on SERVER-STOP and SERVER-OVERLOAD, respectively. These two statistics also struggle to discriminate OoD sequences in LATENCY and SPIKETRAINS scenarios. The non-monotone behavior of the ROC AUC scores for some statistics (as the $\delta$ increases) indicates that these statistics are poorly suited for the respective scenarios.

### 6.3 Detecting anomalies in real-world data

Finally, we apply our methods to detect anomalies in two real-world event sequence datasets. We keep the setup (e.g., configuration of the neural TPP model) identical to Section 6.2.

LOGS: We generate server logs using Sock Shop microservices (Weave, 2017) and represent them as marked event sequences. Sock Shop is a standard testbed for research in microservice applications (Aderaldo et al., 2017) and contains a web application that runs on several containerized services. We generate OoD sequences by injecting various failures (e.g., packet corruption, increased latency) among these microservices using a chaos testing tool Pumba (Ledenev et al., 2016). We split one large server log into 30-second subintervals, that are then partitioned into train and test sets.

STEAD (Stanford Earthquake Dataset) (Mousavi et al., 2019) includes detailed seismic measurements on over 1 million earthquakes. We construct four subsets, each containing 72-hour subintervals in a period of five years within a 350km radius of a fixed geographical location. We treat sequences corresponding the San Mateo, CA region as in-distribution data, and the remaining 3 regions (Anchorage, AK, Aleutian Islands, AK and Helmet, CA) as OoD data.

**Results.** Table 1 shows the ROC AUC scores for all scenarios. KS arrival and chi-squared achieve surprisingly low scores in 6 out of 8 scenarios, even though these two methods showed strong results on simulated data in Sections 6.1 and 6.2. In contrast, KS inter-event and log-likelihood perform better here than in previous experiments, but still produce poor results on Packet corruption. The 3S statistic is the only method that consistently shows high ROC AUC scores across all scenarios. Moreover, we observe that for *marked* sequences (LOGS and all datasets in Section 6.2), the 3S statistic leads to more accurate detection compared to the log-likelihood statistic in 9 out of 9 cases.

## 7 Discussion

**Limitations.** Our approach assumes that the sequences in $\mathcal{D}_{\mathrm{train}}$ were drawn i.i.d. from the true data-generating distribution $\mathbb{P}_{\mathrm{data}}$ (Section 2). This assumption can be violated in two ways: some of the training sequences might be anomalous or there might exist dependencies between them. We have

considered the latter case in our experiments on SPIKETRAINS and LOGS datasets, where despite the non-i.i.d. nature of the data our method was able to accurately detect anomalies. However, there might exist scenarios where the violation of the assumptions significantly degrades the performance.

No single test statistic can be "optimal" for either OoD detection or GoF testing, since we make no assumptions about the alternative distribution $\mathbb{Q}$ (Section 2). We empirically showed that the proposed 3S statistic compares favorably to other choices over a range of datasets and applications domains. Still, for any *fixed* pair of distributions $\mathbb{P}$ and $\mathbb{Q}$, one can always find a statistic that will have equal or higher power s.t. the same false positive rate (Neyman & Pearson, 1933). Hence, it won't be surprising to find cases where our (or any other chosen a priori) statistic is inferior.

**Broader impact.** Continuous-time variable-length event sequences provide a natural representation for data such as electronic health records (Enguehard et al., 2020), server logs (He et al., 2016) and user activity traces (Zhu et al., 2020). The ability to perform unsupervised anomaly detection in such data can enable practitioners to find at-risk patients, reduce DevOps costs, and automatically detect security breaches—all of which are important tasks in the respective fields. One of the risks when applying an anomaly detection method in practice is that the statistical anomalies found by the method will not be relevant for the use case. For example, when looking for health insurance fraud, the method might instead flag legitimate patients who underwent atypically many procedures as "suspicious" and freeze their accounts. To avoid such situations, automated decisions systems should be deployed with care, especially in sensitive domains like healthcare.

**Conclusion.** We have presented an approach for OoD detection for temporal point processes based on goodness-of-fit testing. At the core of our approach lies a new GoF test for standard Poisson processes based on the 3S statistic. Our method applies to a wide class of TPPs and is extremely easy to implement. We empirically showed that the proposed approach leads to better OoD detection accuracy compared to both popular GoF statistics for TPPs (Kolmogorov–Smirnov, chi-squared) and approaches commonly used in OoD detection literature (model log-likelihood). While our analysis focuses on TPPs, we believe our discussion on similarities and distinctions between GoF testing and OoD detection offers insights to the broader machine learning community.

## Funding transparency statement

The work was funded by Amazon Research.

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
