## A Difference between GoF testing and OoD detection

The connection between OoD detection and GoF testing was first pointed out by Nalisnick et al. (2019). They proposed to perform a GoF test for a deep generative model to detect OoD instances. However, as we explained in Section 2, these two problems are in fact *not* equivalent. We now demonstrate how this insight allows us to explain and improve upon some results obtained by Nalisnick et al. (2019).

First, we consider the **Gaussian annulus test** for normalizing flow models that was also used by Choi et al. (2018). A normalizing flow model $\mathbb{P}_{\text{model}}$ defines the distribution of a $D$-dimensional random vector $X$ by specifying a diffeomorphism $f : \mathbb{R}^D \to \mathbb{R}^D$, such that $Z = f(X)$ is distributed according to $\mathcal{N}(\mathbf{0}_D, \mathbf{I}_D)$, the standard normal distribution. In other words, $f(X)|X \sim \mathbb{P}_{\text{model}}$ follows the standard normal distribution, so any test for the normal distribution can be used to test the GoF of a normalizing flow model. Based on this, Nalisnick et al. (2019) define the following test statistic

$$\phi(X) = \left| \|f(X)\|_2 - \mathbb{E}_{X \sim \mathbb{P}_{\text{model}}}[\|f(X)\|_2] \right| = \left| \|f(X)\|_2 - \sqrt{D} \right|. \tag{9}$$

The idea here is to replace a two-sided test on the statistic $\|f(X)\|_2$ with a one-sided test on the statistic $\phi(X)$ defined above. Since $f(X)|X \sim \mathbb{P}_{\text{model}}$ follows the standard normal distribution, the statistic $\phi(X)|H_0$ will concentrate near 0 (Blum et al., 2016, Theorem 2.9). Therefore, checking if $\phi(X)$ is below a certain threshold $\epsilon$ is equivalent to performing the GoF null hypothesis test (Equation 2).

However, the above approach will not work for an OoD detection hypothesis test (Equation 1). If we learn a model $\mathbb{P}_{\text{model}}$ on training instances $\mathcal{D}_{\text{train}}$ that were generated by some distribution $\mathbb{P}_{\text{data}}$, we will in general have $\mathbb{P}_{\text{model}} \neq \mathbb{P}_{\text{data}}$. This implies that $f(X)|X \sim \mathbb{P}_{\text{data}}$ will not follow the standard normal distribution. Therefore, $\mathbb{E}_{X \sim \mathbb{P}_{\text{data}}}[\|f(X)\|_2] \neq \sqrt{D}$ and the distribution of $\|f(X)\|_2$ might not even be symmetric around its mean. This means we cannot replace a two-sided test on $\|f(X)\|_2$ with a one-sided test on $\phi(X)$ when doing OoD detection. A better idea is to directly compute the two-sided $p$-value for the OoD detection test using the statistic $\|f(X)\|_2$, following our approach in Section 2.

Similarly, for the (single-instance) **typicality test**, the test statistic is defined as

$$\gamma(X) = \left| \log q(X) - \mathbb{E}_{X \sim \mathbb{P}_{\text{model}}}[\log q(X)] \right|, \tag{10}$$

where $\log q(X)$ is the log-likelihood of a generative model trained on $\mathcal{D}_{\text{train}}$. This leads to the same problems when trying to apply this statistic for OoD detection as we encountered with the Gaussian annulus test above—the expected value $\mathbb{E}_{X \sim \mathbb{P}_{\text{model}}}[\log q(X)]$ is only suitable for a GoF test. However, in this case Nalisnick et al. (2019) report that they found $\mathbb{E}_{X \sim \mathbb{P}_{\text{data}}}[\log q(X)]$ to work better in practice. By drawing a clear distinction between the OoD detection test and the GoF test we can explain this empirical result. An even better idea is to use the two-sided $p$-value (Equation 3) instead of Equation 10, since the distribution of the statistic $\log q(X)|X \sim \mathbb{P}_{\text{data}}$ is not guaranteed to be symmetric.

## B Other statistics based on squared spacings

The following discussion is based on Moran (1947) and D'Agostino (1986).

**Sum-of-squared spacings (3S) statistic for** $\text{Uniform}([0,1])$**.** Suppose that $\{u_1, \ldots, u_N\}$ are sampled i.i.d. from the $\text{Uniform}([0,1])$ distribution. Additionally, assume w.l.o.g. that the $u_i$'s are sorted in an increasing order, i.e., $u_1 \leq \cdots \leq u_N$. The 3S statistic for $\text{Uniform}([0,1])$ is defined as

$$\psi_N^{\text{Unif}([0,1])} = N \sum_{i=1}^{N+1} (u_i - u_{i-1})^2, \tag{11}$$

where $u_0 = 0$ and $u_{N+1} = 1$. The factor $N$ ensures that $\psi_N^{\text{Unif}([0,1])}$ approaches the standard normal distribution as $N \to \infty$. However, the convergence of $\psi_N^{\text{Unif}([0,1])}$ to its limiting distribution is rather slow.

**3S statistic for** $\text{Uniform}([0, V])$**.** The statistic above can be generalized to the uniform distribution on an arbitrary interval $[0, V]$. Suppose $\{v_1, \ldots, v_N\}$ are drawn i.i.d. from the $\text{Uniform}([0, V])$ distribution, and again are sorted in an increasing order. The 3S statistic for $\text{Uniform}([0, V])$ is defined by simply dividing the $v_i$'s by the interval length $V$.

$$
\begin{aligned}
\psi_N^{\text{Unif}([0,V])} &= N \sum_{i=1}^{N+1} \left( \frac{v_i}{V} - \frac{v_{i-1}}{V} \right)^2 \\
&= \frac{N}{V^2} \sum_{i=1}^{N+1} (v_i - v_{i-1})^2,
\end{aligned}
\tag{12}
$$

where $v_0 = 0$ and $v_{N+1} = V$.

**3S statistic for the SPP on** $[0, V]$**.** Remember that the $N$ factor makes the distribution of $\psi_N^{\text{Unif}([0,V])}$ (asymptotically) invariant for different values of $N$. This means that such statistic wouldn't be able to detect anomalies in terms of the event count. To remove this undesirable property, we define the *3S statistic for the standard Poisson process* by replacing $N$ with its expectation $\mathbb{E}[N|V] = V$.

$$
\begin{aligned}
\psi^{\text{SPP}([0,V])} &= \frac{\mathbb{E}[N|V]}{V^2} \sum_{i=1}^{N+1} (v_i - v_{i-1})^2 \\
&= \frac{1}{V} \sum_{i=1}^{N+1} (v_i - v_{i-1})^2
\end{aligned}
\tag{13}
$$

This is the definition that we introduced in Equation 7. As a side note, replacing $N$ with $\mathbb{E}[N|V]$ is one of the possible choices that ensures that (1) the statistic is sensitive to changes in the event count and (2) the expected value $\mathbb{E}[\psi^{\text{SPP}([0,V])}|V]$ doesn't change for different values of $V$, and hence is comparable across different transformed sequences.

As we show in Sections 4 and 6, the above definition of the 3S statistic for the SPP allows us to detect a broad class of anomalies (i.e., deviations from the SPP) that differ both in the distribution of the event count $N$ as well as the arrival times $v_i$.

## C   Proof of Proposition 1

To compute the moments of the 3S statistic for the standard Poisson process (Equation 7) we need to marginalize out the event count $N$, which is equivalent to applying the law of iterated expectation

$$
\begin{aligned}
\mathbb{E}[f(\psi)|T] &= \sum_{n=0}^{\infty} \mathbb{E}[f(\psi)|N = n, V] \Pr(N = n|V) \\
&= \sum_{n=0}^{\infty} \mathbb{E}[f(\psi)|N = n, V] \frac{V^n e^{-V}}{n!}
\end{aligned}
\tag{14}
$$

where we used the fact that $N|V \sim \text{Poisson}(V)$.

We obtain the expectations of $\psi$ and $\psi^2$ conditioned on $N$ and $V$ using the result by Moran (1947) on the moments of $\psi_N^{\text{Unif}([0,1])}$ (Equation 11), the 3S statistic for the $\text{Uniform}([0, 1])$ distribution.

$$
\begin{aligned}
\mathbb{E}[\psi|N = n, V] &= \frac{2V}{(n + 2)} \\
\mathbb{E}[\psi^2|N = n, V] &= \frac{4V^2(n + 6)}{(n + 2)(n + 3)(n + 4)}
\end{aligned}
\tag{15}
$$

These can also be easily derived from the moments of the Dirichlet distribution, by using the fact that the scaled inter-event times $(w_1/V, \ldots, w_{N+1}/V)$ are distributed uniformly on the standard $N$-simplex (i.e., according to Dirichlet distribution with parameter $\boldsymbol{\alpha} = \mathbf{1}_{N+1}$).

By plugging in Equation 15 into Equation 14, we obtain

$$
\begin{aligned}
\mathbb{E}[\psi|V] &= 2Ve^{-V} \sum_{n=0}^{\infty} \frac{V^n}{n!(n+2)} \\
&= 2Ve^{-V} \frac{1}{V^2} \left( e^V(V-1) + 1 \right) \\
&= \frac{2}{V}(V + e^{-V} - 1).
\end{aligned}
\tag{16}
$$

Similarly, we compute the non-centered second moment as

$$
\begin{aligned}
\mathbb{E}[\psi^2|V] &= 4V^2 e^{-V} \sum_{n=0}^{\infty} \frac{V^n(n+6)}{n!(n+2)(n+3)(n+4)} \\
&= 4V^2 e^{-V} \frac{1}{V^4} \left( e^V(V^2-6) + 2(V^2 + 3V + 3) \right) \\
&= \frac{4}{V^2} \left( V^2 - 6 + 2e^{-V}(V^2 + 3V + 3) \right).
\end{aligned}
\tag{17}
$$

Finally, we obtain the variance as

$$
\begin{aligned}
\mathrm{Var}[\psi|V] &= \mathbb{E}[\psi^2|V] - \mathbb{E}[\psi|V]^2 \\
&= \frac{4}{V^2} \left( 2V - 7 + e^{-V}(2V^2 + 4V + 8 - e^{-V}) \right).
\end{aligned}
$$

Higher-order moments of $\psi|V$ can be computed similarly using Equation 14.

## D   Implementation details

The following code describes the procedure for computing the $p$-values for both hypothesis tests discussed in Section 2—namely, the GoF test (Equation 2) and the OoD detection test (Equation 1). The code below is for demonstration purposes only, the actual implementation used in our experiments is better optimized.

```
def compute_p_value(x_test, samples, score_fn):
    scores_id = [score_fn(x) for x in samples]
    score_x = score_fn(x_test)
    num_train = len(samples)
    num_above = 0
    for s in scores_id:
        if s > score_x:
            num_above += 1
    num_below = num_train - num_above
    return min(
        (num_below + 1) / (num_train + 1),
        (num_above + 1) / (num_train + 1)
    )
```

The +1 correction in the numerator and denominator for the $p$-value computation is done as described by North et al. (2002). If we define `samples` as the set of in-distribution sequences $\mathcal{D}_{\text{train}}$ that were generated from $\mathbb{P}_{\text{data}}$, we recover the OoD detection test (Equation 1 and Section 4.2). If we define `samples` as the set of sequences $\mathcal{D}_{\text{model}}$ that were generated from $\mathbb{P}_{\text{model}}$, we recover the GoF test (Equation 2 and Section 4.1).

In the snippet above, `score_fn` corresponds to a test statistic $s\colon \mathcal{X} \to \mathbb{R}$. In our experiments, we consider the following choices for $s$:

1. KS arrival (Equation 5).

2. KS inter-event (Equation 6).

3. Chi-squared: we partition the interval $[0, V]$ into $B = 10$ disjoint buckets of equal length, and compare the observed event count $N_b$ in each bucket with the expected amount $L = V/B$

$$
\chi^2(Z) = \sum_{b=1}^{B} \frac{(N_b - L)^2}{L}.
\tag{18}
$$

4. Sum-of-squared spacings (Equation 7).

5. Log-likelihood

$$\log q(X) = \sum_{i=1}^{N} \log \frac{\partial \Lambda^*(t_i)}{\partial t_i} - \Lambda^*(T). \tag{19}$$

All these statistics are computed based on some TPP model with compensator $\Lambda^*$. For statistic 1–4, we compute $s(X)$ by first obtaining the transformed sequence $Z = (\Lambda^*(t_1), \ldots, \Lambda^*(T))$ and then evaluating the respective SPP statistic on $Z$. The log-likelihood is directly evaluated based on the model's conditional intensity.

**Marked sequences.** In a marked sequence $X = \{(t_1, m_1), \ldots, (t_N, m_N)\}$ each event is represented by a categorical mark $m_i \in \{1, \ldots, K\}$ in addition to the arrival time $t_i$. A marked TPP model is specified by $K$ compensators $\{\Lambda_1^*, \ldots, \Lambda_K^*\}$.

We obtain the transformed sequence $Z$ necessary for statistics 1–4 as follows. Let $\left( t_1^{(k)}, \ldots, t_{N_k}^{(k)} \right)$ denote the events of mark $k$ in a given sequence $X$. For each mark $k \in \{1, \ldots, K\}$, we obtain a transformed sequence $Z^{(k)} = \left( \Lambda_k^*(t_1^{(k)}), \ldots, \Lambda_k^*(t_{N_k}^{(k)}), \Lambda_k^*(T) \right)$. Then we concatenate the transformed sequences for each mark, thus obtaining a single SPP realization on the interval $[0, \sum_{k=1}^{K} \Lambda_k^*(T)]$. For example, suppose the transformed sequence for the first mark is $Z^{(1)} = (1.0, 2.5, 4.0)$ and for the second mark $Z^{(2)} = (0.5, 3.0)$. Then the concatenated sequence will be $Z = (0.0, 1.0, 2.5, 4.0 + 0.5, 4.0 + 3.0) = (0.0, 1.0, 2.5, 4.5, 7.0)$. Our approach based on concatenating the $Z^{(k)}$'s is simpler than other methods for combining multiple sequences by Gerhard et al. (2011) & Tao et al. (2018), and we found ours to work well in practice.

The log-likelihood for a marked sequence is computed as

$$\log q(X) = \sum_{k=1}^{K} \sum_{i=1}^{N} \mathbb{1}(m_i = k) \log \frac{\partial \Lambda_k^*(t_i)}{\partial t_i} - \sum_{k=1}^{K} \Lambda_k^*(T). \tag{20}$$

## E  Datasets

### E.1  Standard Poisson process

In-distribution sequences (corresponding to $\mathbb{P}_{\text{model}}$) are all generated from an SPP (i.e., a homogeneous Poisson process with rate $\mu = 1$) on the interval $[0, 100]$. The OoD sequences (corresponding to $\mathbb{Q}$) for each of the scenarios are generated as follows, where $\delta \in [0, 1]$ is the detectability parameter.

(i) RATE: homogeneous Poisson process with rate $\mu = 1 - 0.5\delta$.

(ii) STOPPING: We generate a sequence $X = (t_1, \ldots, t_N)$ from an SPP and then remove all the events $t_i \in [t_{\text{stop}}, T]$, where we compute $t_{\text{stop}} = T(1 - 0.3\delta)$.

(iii) RENEWAL: A renewal process, where the inter-event times $\tau_i$ are sampled i.i.d. from a Gamma distribution with shape $k = 1 - \delta$ and scale $\theta = \frac{1}{1-\delta}$. Thus, the expected inter-event time stays the same, but the variance of inter-event times increases for higher $\delta$.

(iv) HAWKES: Hawkes process with conditional intensity $\lambda^*(t) = \mu + \alpha \sum_{t_j < t} \exp(-(t - t_j))$. The parameters are chosen as $\mu = 1 - \delta$ and $\alpha = \delta$.

(v) INHOMOGENEOUS: inhomogeneous Poisson process with intensity $\lambda(t) = 1 + \beta \sin(\omega t)$, where $\omega = \frac{2\pi}{50}$ and $\beta = 2\delta$.

(vi) SELFCORRECTING: self-correcting process with intensity $\lambda^*(t) = \exp \left( \mu t - \sum_{t_j < t} \alpha \right)$, where we set $\mu = \delta + 10^{-5}$ and $\alpha = \delta$.

In all above scenarios setting $\delta = 0$ recovers the standard Poisson process, thus making $\mathbb{P}_{\text{data}}$ and $\mathbb{Q}$ indistinguishable. Note that the parameters in scenarios (iii)–(vi) are chosen such that the expected number of events $N$ is always equal to $T$, like in the SPP. For all scenarios, $\mathcal{D}_{\text{train}}$, $\mathcal{D}_{\text{test}}^{\text{ID}}$ and $\mathcal{D}_{\text{test}}^{\text{OOD}}$ consist of 1000 sequences each.

**Additional experiments.** For completeness, we consider two more scenarios.

(vii) INCREASINGRATE: Similar to scenario (i), but now the rate is increasing instead as $\mu = 1 + 0.5\delta$.

(viii) RENEWALB: Similar to scenario (iii), but the variance now decreases for higher $\delta$. For this we define the parameters of the Gamma distribution as $k = \frac{1}{1-\delta}$ and $\theta = 1 - \delta$.

The results are shown in Figure 6. As we can see, the same qualitative conclusions apply here as for the experiments in Section 6.1.

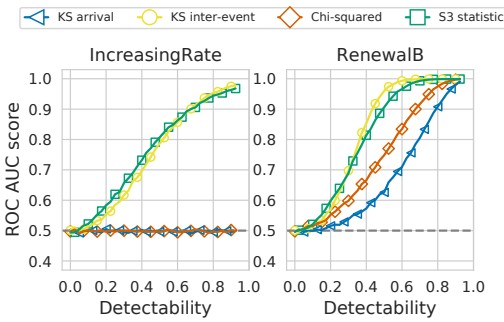

Figure 6: GoF testing for the SPP using additional scenarios.

## E.2   Simulated data

SERVER-STOP and SERVER-OVERLOAD: In-distribution sequences for both scenarios are generated by a multivariate Hawkes process with $K = 3$ marks on the interval $[0, 100]$ with following base rates $\boldsymbol{\mu}$ and influence matrix $\boldsymbol{A}$:

$$\boldsymbol{\mu} = \begin{pmatrix} 3 \\ 0 \\ 0 \end{pmatrix} \qquad\qquad \boldsymbol{A} = \begin{pmatrix} 0 & 0 & 0 \\ 1 & 0 & 0 \\ 1 & 0 & 0 \end{pmatrix}$$

This scenario represents communication between a server (mark 1) and two worker machines (marks 2 and 3)—events for the workers can only be triggered by incoming requests from the server.

In OoD sequences, the structure of the influence matrix is changed at time $t_{\text{stop}} = T(1 - 0.5\delta)$, which represents the time of a failure in the system. For SERVER-STOP, the influence matrix is changed to $\boldsymbol{A}^{\text{stop}}$, and for SERVER-OVERLOAD the influence matrix is changed to $\boldsymbol{A}^{\text{overload}}$.

$$\boldsymbol{A}^{\text{stop}} = \begin{pmatrix} 0 & 0 & 0 \\ 0 & 0 & 0 \\ 1 & 0 & 0 \end{pmatrix} \qquad\qquad \boldsymbol{A}^{\text{overload}} = \begin{pmatrix} 0 & 0 & 0 \\ 0 & 0 & 0 \\ 2 & 0 & 0 \end{pmatrix}$$

The sets $\mathcal{D}_{\text{train}}$, $\mathcal{D}_{\text{test}}^{\text{ID}}$ and $\mathcal{D}_{\text{test}}^{\text{OOD}}$ consist of 1000 sequences each.

LATENCY: Event sequences consist of two marks. ID sequences are generated as follows. Events of the first mark ("the trigger") are generated by a homogeneous Poisson process with rate $\mu = 3$. Events of the the second mark ("the response") are obtained by shifting the arrival times of the first mark by offsets that are sampled i.i.d. from $\text{Normal}(\mu = 1, \sigma = 0.1)$. In OoD sequences, the offsets are instead sampled from $\text{Normal}(\mu = 1 + 0.5\delta, \sigma = 0.1)$. That is, OoD sequences correspond to increased latency between the "trigger" and "response" events. The sets $\mathcal{D}_{\text{train}}$, $\mathcal{D}_{\text{test}}^{\text{ID}}$ and $\mathcal{D}_{\text{test}}^{\text{OOD}}$ consist of 1000 sequences each.

SPIKETRAINS: The original fluorescence data is provided at www.kaggle.com/c/connectomics. We extracted the spike times from the fluorescence recordings using the code by https://github.com/slinderman/pyhawkes/tree/master/data/chalearn. We dequantized the discrete spike times by adding $\text{Uniform}(-0.5, 0.5)$ noise and selected the first 50 marks.

The original data consists of a single sequence that is 3590 seconds long. We split the long sequence into overlapping windows that are 20 seconds long. We select the first 500 sequences for training (as $\mathcal{D}_{\text{train}}$), and 96 remaining sequences for testing (as $\mathcal{D}_{\text{test}}^{\text{ID}}$). OoD sequences (i.e., $\mathcal{D}_{\text{test}}^{\text{OOD}}$) are obtained by switching $k = \lfloor \delta K \rfloor$ marks. For example, if marks 5 and 10 are switched, all events that correspond to mark 5 in $\mathcal{D}_{\text{test}}^{\text{ID}}$ will be labeled as mark 10 in $\mathcal{D}_{\text{test}}^{\text{OOD}}$, and vice versa.

### E.3 Real-world data

LOGS: We ran the Sock Shop microservices testbed (Weave, 2017) on our in-house server. We consider the logs corresponding to the `user` service. There are 4 types of log entries that we model as 4 categorical marks. We use the timestamps of log entries as arrival times of a TPP. We slice the logs into 30-second-long non-overlapping windows, each corresponding to a single TPP realization.

We run the service for ≈14 hours to generate training data, and then for additional ≈5 hours to generate test data. The test data contains 5 types of injected anomalies produced by Pumba (Ledenev et al., 2016). See Table 1 for the list of anomalies. Each anomaly injection lasts 10 minutes. We mark a test sequence as OoD if the system was "attacked" by Pumba during the respective time window. In total, we use 1668 sequences as $\mathcal{D}_{\text{train}}$, 502 sequences as $\mathcal{D}_{\text{test}}^{\text{ID}}$, and 22 sequences as $\mathcal{D}_{\text{test}}^{\text{OOD}}$ for each of the attack scenarios (i.e., 110 OoD sequences in total).

STEAD: The original dataset by Mousavi et al. (2019) contains over 1 million earthquake recordings. We sample 72-hour sub-windows and treat times of earthquake as arrival times of a TPP, as usually done in seismological applications. We treat the sequences as unmarked. We select 4 geographic locations: (1) San Mateo, CA, (2) Anchorage, AK, (3) Aleutian Islands, AK, and (4) Hemet, CA. We group the earthquakes that happen within a 350 km radius (geodesic) around each of the locations, thus obtaining 4 sets of sequences (5000 sequences for each location). We use the sequences corresponding to (1) San Mateo, CA, as in-distribution data, and the remaining 3 locations as OoD data. We use 4000 ID sequences as $\mathcal{D}_{\text{train}}$, 1000 ID sequences and $\mathcal{D}_{\text{test}}^{\text{ID}}$, and 1000 sequences per each remaining location as $\mathcal{D}_{\text{test}}^{\text{OOD}}$.

## F  Experimental setup

### F.1  GoF for the SPP (Section 6.1)

We compute the $p$-values for the GoF test using the procedure described in Appendix D. For the GoF test, we use 1000 event sequences generated by an SPP as `samples` in the algorithm. The test statistics are computed using the compensator of the SPP $\Lambda^*(t) = t$. We compute the $p$-value for each event sequence in $\mathcal{D}_{\text{test}}^{\text{ID}}$ and $\mathcal{D}_{\text{test}}^{\text{OOD}}$, and then compute the ROC AUC score based on these $p$-values. The results are averaged over 10 random seeds.

### F.2  OoD detection (Sections 6.2 & 6.3)

We train a neural TPP model similar to Shchur et al. (2020). We parametrize the inter-event time distribution with a mixture of 8 Weibull distributions. The marks are conditionally independent of the inter-event times given the context embedding, as in the original model. Mark embedding size is set to 32, and the context embedding (i.e., RNN hidden size) is set to 64 for all experiments.

We optimize the model parameters by maximizing the log-likelihood of the sequences in $\mathcal{D}_{\text{train}}$ (batch size 64) using Adam with learning rate $10^{-3}$ and clipping the $L_2$-norm of the gradients to 5. We run the optimization procedure for up to 200 epochs, and perform early stopping if the training loss stops improving for 10 epochs. The $p$-values are computed according to the procedure described in Appendix D. The results reported in Section 6.2 are averaged over 10 random seeds. In Section 6.3, we train the neural TPP model with 5 different random initializations to compute the average and standard error in Table 1.

## G  Fisher's method for KS statistics

Here we show that ad-hoc fixes to the KS statistics that make them sensitive to the variations in the event count $N$ lead to worse discriminative power in other scenarios. For this, we replicate the experimental setup from Section 6.1 with two additional statistics.

**Fisher arrival.** We compute the two-sided $p$-value $p_N$ for the event count $N$ using the CDF of the $\text{Poisson}(V)$ distribution. Then, we compute the two-sided $p$-value $p_{\kappa_{\text{arr}}}$ for KS arrival statistic (Equation 5) using Kolmogorov distribution. We combine the two $p$-values using Fisher's method (Fisher, 1948) as $-2(\log p_N + \log p_{\kappa_{\text{arr}}})$. **Fisher inter-event** is defined similarly using the two-sided $p$-value $p_{\kappa_{\text{int}}}$ for the KS inter-event statistic (Equation 6).

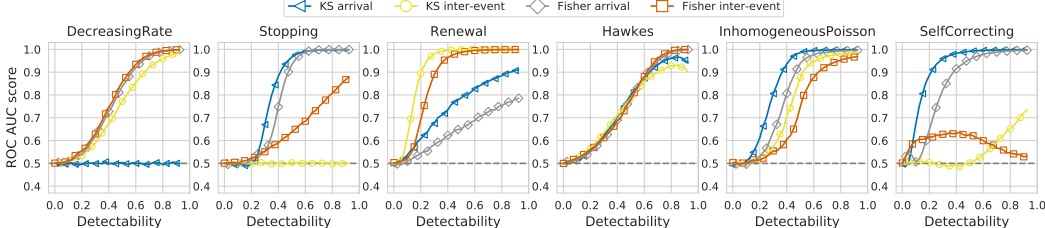

Figure 7: Comparing KS statistics with the respective Fisher versions that are sensitive to the event count $N$.

**Results.** Results are shown in Figure 7. We see that the Fisher versions of the statistics indeed fix the failure modes of the two KS scores on RATE and STOPPING, where the event count $N$ changes in OoD sequences. However, the Fisher versions of the statistics perform worse than the respective KS statistics in 3 out of 4 remaining scenarios. In contrast, the 3S statistic performs well both in scenarios where $N$ changes, as well as when the distribution of the arrival/inter-event times is varied.