# OpenReview forum: "Detecting Anomalous Event Sequences with Temporal Point Processes"
_NeurIPS.cc/2021/Conference — NeurIPS 2021 Poster_

### Official Review · Reviewer_XhWV · 2021-07-13

**Rating:** 7
**Confidence:** 2

**Summary:**

This paper focuses on the anomaly detection of continuous-time event sequences with temporal point processes and proposes to leverage the idea of goodness-of-fit testing to check whether sequences follow the in-distribution. Consider the limitation that the current statistics are not sensitive to the event number N, this paper proposes a new statistic called sum-of-squared-spacings (3S) to check the fitness of the sequence. The experimental results show the effectiveness of the proposed statistic for anomaly detection.

**Limitations And Societal Impact:**

Two minor comments here. First, I personally would expect more explanation about the advantages of the proposed 3S statistic over the log-likelihood approach. As mentioned in the paper, many existing anomaly detection approaches are actually based on log-likelihood, because the log-likelihood approach is more intuitive. The experiments on the real-world datasets in this paper also show that log-likelihood can achieve good performance. I believe give more comparisons about these two approaches would be appreciated. Another comment may be outside the scope of this paper, but I am curious about how to detect the marked event sequences with the marked TPP model. Say, using log-likelihood, it would be easy to combine the signals about marked sequences as well as the time sequences for anomaly detection because both can be modeled via log-likelihood, but the proposed 3S statistic can only detect anomalous sequences from the time perspective.

**Main Review:**

Detecting anomalous continuous-time event sequences is an important task yet under-exploited. This paper proposes a new statistic to identify the anomalous event sequences, called 3S. Evaluating on two synthetic and one real datasets shows the effectiveness of using 3S for anomaly detection. The overall paper is well-written with sufficient references and experiments.

**Time Spent Reviewing:**

3

---

> ### Author Response · Authors · 2021-08-09
> **Authors' response**
>
> Thank you for your feedback and for the interesting questions!
>
> The approaches based on the random time change theorem (i.e., 3S, KS-arrival, KS-interevent and chi-squared statistics) can be applied to sequences with categorical marks. For this we can simply concatenate the transformed sequence for each mark into a single sequence and compute the statistic on it (see Appendix D for details). This is theoretically justified for GoF testing due to the independence property of the Poisson process. All simulated datasets in Section 6.2 and the ServerLogs dataset in Section 6.3 contain marks (# of marks ranging from 2 to 50), and our proposed statistic works better than log-likelihood in 9 out of 9 cases. All these details are currently mostly described in the appendix, but we will make them more prominent in the main paper in the revised version.
>
> Regarding your first question, there may exist scenarios where the log-likelihood (LL) statistic is superior to our proposed approach. One way to summarize the message of our paper is as follows: “While most existing OoD detection approaches focus on LL, we show that GoF statistics provide a promising alternative that in many cases leads to more accurate detection of anomalies”. Moreover, we show some failure modes of the LL statistic in Section 6.2. We will discuss these failure modes in more detail and explain how our approach positions w.r.t. LL-based approaches in the updated version.

---

### Official Review · Reviewer_jvJp · 2021-07-15

**Rating:** 7
**Confidence:** 3

**Summary:**

This paper discusses anomaly detection in continuous-time event sequences as out-of-distribution (OoD) detection for temporal point processes, and statistical tests for the detection. The paper proposes a new statistical test for the detection and demonstrates the usefulness of the proposed test in simulated and real-world data sets.


**Limitations And Societal Impact:**

See above.

**Main Review:**

This is a well-written paper. The connection between anomaly detection in continuous-time event sequences and out of distribution detection has been discussed, and such a connection is useful for anomaly detection research. The proposed 3S test is well motivated and sound. 3S test has been shown effective for anomaly detection in continuous-time event sequences in experiments.

Discussions in the paper are focused, clear and easy to follow. The authors evaluate the proposed test in different circumstances, and the strengths and weaknesses of the proposed test have been presented. The connection between anomaly detection and out of distribution detection, and existing and proposed statistical tests are useful for anomaly detection research.


**Time Spent Reviewing:**

3

---

> ### Author Response · Authors · 2021-08-09
> **Authors' response**
>
> Thank you for your encouraging feedback! Based on your feedback and that from the other reviewers, we will make some changes to the paper, in particular adding some further details and discussion on the 3S test.

---

### Official Review · Reviewer_kuGg · 2021-07-17

**Rating:** 7
**Confidence:** 4

**Summary:**

This paper addresses an anomaly detection problem for event sequences as temporal point process (TPP), where they formalize the problem as out-of-distribution (OoD) detection in a batch scenario.
The authors proposed using goodness-of-fit (GoF) statistics to determine whether a sequence is OoD or not, which provides vast options for this purpose to us than an ordinary OoD procedure just using log-likelihood.
They make it possible to use GoF statistics designed for known standard Poisson process by using Neural TPP fitted to training sequences. We can convert a random event sequence distributed according to an arbitrary TPP into a realization of the standard Poisson process with a compensator derived from the fitted Neural TPP.
They also propose a sum-of-squared-spacings (3S) statistic that can detect anomalies w.r.t. the event count N, which is difficult to detect by Kolmogorov–Smirnov (KS) statistic, as well as anomalies w.r.t. the distribution of the arrivals or inter-event times.
Experimental results demonstrate that the proposed statistic performs better than either other statistics or log-likelihood-based OoD.

**Limitations And Societal Impact:**

They addressed the limitations and potential negative social impact of their work.

**Main Review:**

### Originality:
* This work proposes a novel combination of GoF statistic and OoD for event sequences with learned neural TPP for the unknown data-generating distribution.


### Quality:
* The authors introduced the motivation of the proposed method and provided the necessary background, which would be informative for many readers.

* The requirements of the proposed method would limit its practical usability.
    - $T$ number of observations are required for anomaly detection with the proposed method. Also, $T$ and $V$ are assumed to be large enough in their analysis. In that case, it is challenging to perform timely anomaly detection, which is crucial for real applications, such as detecting intrusions in computer networks and detecting fraud or shifts in the market structure on financial systems. It would be better to add analysis w.r.t different configurations for $V$.

* The motivation and justification of the use of 3S statistics are unclear.
    - Is it possible to use other statistics instead of 3S statistics?
    - In l.562 in Appendix, the authors replaced $N$ with its expectation $E[N |V ]$. I cannot find the justification for that. Is it just heuristics for considering $N$ as a random variable? Are there any other variants?

* The compared methods seem naive.
    - They analyze when KS statistic is modified by removing $\sqrt{N}$, but they can also replace $1/N$ in the second equation with $1/V$ similar to the proposed method. It would work better.
    - It would be better to add a comparison with the original 3S statistic without modification.


### Clarity:
* The discussion in l.195 is confusing since it would be mainly for KS statistic, not for 3S statistic, which does not require CDF. If so, the paragraph from l.195 would not be a help to understand the proposed method.

* There is no description of how to choose hyperparameters of models, such as how they split validation set from the training set and test set.

* In section 6.3, there is no description of the model they used.

* Figure 1 is not mentioned in the main text. Also, in general, it is better to locate figures on top.


### Significance:
* The practical usability of the proposed method can be limited because of its requirements, as stated above.
* Since the compared methods seem naive in the experiments, it is not easy to find the significance of the proposed method.


================ Update: The motivation and novelty of the proposed method have been clarified in the rebuttal. Also, contributions on empirical evaluation should be noted, where the proposed approach worked well even on short intervals.

**Time Spent Reviewing:**

7

---

> ### Author Response · Authors · 2021-08-09
> **Authors' response**
>
> Thank you for your feedback and practical suggestions for improving our paper! We would like to address some of the concerns raised in your review:
> 1. **Requirements of the method:** While the analysis in Section 4.1 considers the case where the interval length $V$ is high, our experiments in Section 6 demonstrate that in practice our approach works well even on short intervals. Specifically, the method excels on the Server Logs dataset consists of 30-second-long time intervals (Table 1). On simulated data, we can perfectly detect OoD subsequences based on the 3S statistic even when only 5% of the interval are affected by anomalies (Figure 5, scenarios ServerStop, ServerOverload). We will highlight this aspect in the revised version of the paper.
> 2. **Motivation for the 3S statistic & other statistics**: We would like to highlight that our usage of GoF statistics to detect OoD instances is itself novel. Therefore, the other statistics (Kolmogorov-Smirnov, Chi-squared) are not “baselines” in the usual meaning of the word, but rather other instantiations of our proposed framework. Our contributions (lines 31-35) are showing that (1) GoF statistics can be used to perform OoD/anomaly detection and (2) that the 3S statistic works well for this purpose in a range of scenarios due to its attractive properties. As we discussed in Section 7 (under Limitations), no single statistic can be “optimal” unless we make assumptions about the distributions $\mathbb{P}$ and $\mathbb{Q}$. Therefore, combining our framework with other statistics might lead to better results in certain scenarios. We will make this discussion more prominent and reiterate this point earlier in the paper.
> 3. **Definition of the 3S statistic**: Replacing $N$ with $\mathbb{E}[N|V]$ is one possible choice that ensures that (1) the statistic is sensitive to changes in the event count and (2) the expected value of the statistic $\mathbb{E}[\phi|V]$ (almost) doesn’t change for different values of $V$. Keeping the original version with $N$ would make the statistic (asymptotically) invariant with respect to $N$, and thus will prevent us from detecting anomalies in the event count, which we have shown to be very important in Section 6.1. We will clarify this in the paper---thank you for the suggestion.
> 4. **Compared methods**: We considered all the widely used GoF statistics for TPPs in our comparison (see references in Sections 3 & 5). The $\frac{1}{N}$ factor in Equation 5 comes from the definition of the empirical CDF, so changing it seems to completely alter the interpretation of the KS statistic. As mentioned in point #2 above, it is possible that combining our framework with other statistics (or modifying existing ones) would lead to better results in some cases. This, however, doesn’t invalidate the contributions of our work. Finally, we would like to point out that we avoided tweaking the definition of the statistics after computing the results on the test set to ensure a fair comparison.
> 5. **Experimental setup**: We used the same neural TPP model for the experiments in both Sections 6.2 and 6.3. The hyperparameters of the model were chosen similar to the original paper that proposes this model and were fixed across all experiments. We didn’t use a validation set: the model was trained until convergence on the training set, and the hyperparameters were fixed a priori. Currently, this discussion is mostly contained in Appendix F.2, but we will move it to the main paper.
> 6. **Discussion in line 195**: We need to compute (or approximate) the CDF of the test statistic under $H_0$ to compute the p-value (Equation 3). This applies to all test statistics, including 3S and KS. We highlight the case with the KS statistic since it is instructive for understanding some of the pitfalls of prior works (Appendix C). We will improve this discussion in the revised version.

---

> > ### Comment · Reviewer_kuGg · 2021-08-29
> > **Thank you for your detailed response.**
> >
> > I feel that you have addressed my comments well, especially on motivation and clarity, which were my primary concerns.
> > Also, I admit that contributions on empirical evaluation should be noted, where the proposed approach worked well even on short intervals.
> > I raised my score to a 7.

---

### Decision · Program_Chairs · 2021-09-27

**Decision:**

Accept (Poster)

**Comment:**

The authors formulate anomaly detection in continuous-time event sequences as an out-of-distribution detection problem for temporal point processes.  This allows them to apply a number of common goodness-of-fit measures to detect the out-of-distribution data (e.g., KS tests on arrival or inter-event times), and propose other statistics (specifically, 3S or "sum of squared spacings") and test for typicality of the statistic values under the training dataset.  Overall reviewers found the work clear and novel and recommend acceptance.  Please see the reviews for more detailed suggestions to improve the presentation.